# Across-species differences in pitch perception are consistent with differences in cochlear filtering

**Kerry MM Walker[1]\*, Ray Gonzalez[2], Joe Z Kang[1], Josh H McDermott[2,3][†], Andrew J King[1][†]**

[1]Department of Physiology, Anatomy & Genetics, University of Oxford, Oxford, United Kingdom; [2]Department of Brain and Cognitive Sciences, Massachusetts Institute of Technology, Cambridge, United States; [3]Program in Speech and Hearing Biosciences and Technology, Harvard University, Cambridge, United States

**Abstract** Pitch perception is critical for recognizing speech, music and animal vocalizations, but its neurobiological basis remains unsettled, in part because of divergent results across species. We investigated whether species-specific differences exist in the cues used to perceive pitch and whether these can be accounted for by differences in the auditory periphery. Ferrets accurately generalized pitch discriminations to untrained stimuli whenever temporal envelope cues were robust in the probe sounds, but not when resolved harmonics were the main available cue. By contrast, human listeners exhibited the opposite pattern of results on an analogous task, consistent with previous studies. Simulated cochlear responses in the two species suggest that differences in the relative salience of the two pitch cues can be attributed to differences in cochlear filter bandwidths. The results support the view that cross-species variation in pitch perception reflects the constraints of estimating a sound's fundamental frequency given species-specific cochlear tuning.
DOI: https://doi.org/10.7554/eLife.41626.001

**\*For correspondence:**
kerry.walker@dpag.ox.ac.uk

[†]These authors contributed equally to this work

## Introduction

Many of the sounds in our environment are periodic, and the rate at which such sounds repeat is known as their fundamental frequency, or F0. We perceive the F0 of a sound as its pitch, and this tonal quality is one of the most important features of our listening experience. The way that F0 changes encodes meaning in speech (*Ohala, 1983*) and musical melody (*Cousineau et al., 2009*; *Dowling and Fujitani, 1971*; *Krumhansl, 1990*). The F0 of a person's voice provides a cue to their identity (*Gelfer and Mikos, 2005*; *Latinus and Belin, 2011*; *McPherson and McDermott, 2018*) and helps us attend to them in a noisy environment (*Darwin, 2005*; *Miller et al., 2010*; *Popham et al., 2018*).

The vocal calls of non-human animals are also often periodic, and pitch is believed to help them to identify individuals and interpret communication calls (*Koda and Masataka, 2002*; *Nelson, 1989*). Many mammalian species have been shown to discriminate the F0 of periodic sounds in experimental settings (*Heffner and Whitfield, 1976*; *Osmanski et al., 2013*; *Shofner et al., 2007*; *Tomlinson and Schwarz, 1988*; *Walker et al., 2009*), and these animal models hold promise for understanding the neural mechanisms that underlie pitch perception. However, pitch acuity can differ markedly across species (*Shofner, 2005*; *Walker et al., 2009*), raising the possibility that humans and other mammals may use different neural mechanisms to extract pitch.

The auditory cortex plays a key role in pitch processing, but it remains unclear how cortical neurons extract the F0 of a sound (*Wang and Walker, 2012*). Neural correlates of F0 cues (*Bendor and*

Wang, 2010; Bizley et al., 2009; Fishman et al., 2013) and pitch judgments (Bizley et al., 2013) have been observed across auditory cortical fields in some species, while specialized 'pitch-tuned' neurons have thus far only been described in marmoset auditory cortex (Bendor and Wang, 2005). There is a similar lack of consensus regarding the neural code for pitch in the human brain (Griffiths and Hall, 2012). A better understanding of the similarities and differences in pitch processing across species is essential for interpreting neurophysiological results in animals and relating them to human pitch perception.

Pitch discrimination in humans is thought to be driven by two acoustical cues that result from low-numbered 'resolved' harmonics and high-numbered 'unresolved' harmonics (Moore and Gockel, 2011). The relative importance of these cues offers a means to compare pitch mechanisms across species. In the frequency domain, F0 can be determined from the distribution of harmonics (Figure 1A, upper panel) (Goldstein, 1973; Shamma and Klein, 2000; Terhardt, 1974). In the auditory nerve, the frequency spectrum is represented as a 'place code' of activation across the tonotopic map as well as a 'temporal code' of spikes that are phase-locked to basilar membrane vibrations (Cariani and Delgutte, 1996; Schnupp et al., 2011). However, both these representations are limited by the cochlea's frequency resolution (Goldstein, 1973). Because cochlear filter bandwidths increase with frequency, only low-numbered harmonics produce discernible peaks of excitation and phase locked spikes at their centre frequency (Figure 1A, middle panel). Such harmonics are said to be 'resolved'. By contrast, high-numbered harmonics are not individually resolved, and instead produce beating in time at the F0, conveyed by phase-locking to their envelope (Schouten, 1970) (Figure 1A, bottom panel). For convenience and to be consistent with prior literature, we refer to these unresolved pitch cues as 'temporal' cues, cognizant that the representation of resolved harmonics may also derive from a temporal neural code.

Although psychophysical experiments have demonstrated that humans can extract F0 using either resolved harmonics or unresolved harmonics alone (Bernstein and Oxenham, 2003; Houtsma and Smurzynski, 1990; Shackleton and Carlyon, 1994), human pitch perception is generally dominated by resolved harmonics (Ritsma, 1967; Shackleton and Carlyon, 1994; Shofner and Campbell, 2012). Marmosets can also use resolved harmonics to detect F0 changes (Bendor et al., 2012; Song et al., 2016), whereas rodents (i.e. gerbils and chinchillas) rely more upon temporal periodicity cues (Klinge and Klump, 2010; Klinge and Klump, 2009; Shofner and Chaney, 2013). It has been suggested that these apparent species differences in perception could relate to the pitch cues that are available following cochlear filtering (Cedolin and Delgutte, 2010; Shofner and Chaney, 2013). In particular, the growing evidence that cochlear bandwidths are broader in many non-human species (Joris et al., 2011; Shera et al., 2002), including ferrets (Alves-Pinto et al., 2016; Sumner et al., 2018), supports the possibility that they might process pitch cues in different ways from humans, as has been noted previously (Shofner and Campbell, 2012; Shofner and Chaney, 2013).

The behavioural studies carried out to date are difficult to compare across species. First, pitch in humans is defined as the percept through which sounds are ordered on a scale from low to high (ANSI-S1, 2013). By contrast, animal studies often measure change detection on a go/no-go task, from which it is difficult to determine whether they experience a change in a comparably ordered pitch percept rather than some aspect of timbre. A two-alternative forced choice (2AFC) task requiring 'low' and 'high' judgements analogous to those used in human psychophysical tasks would better enable cross-species comparisons (Walker et al., 2009), but has yet to be employed to examine the use of resolved and unresolved cues in animals. Second, the spectral range of stimuli was not fully controlled across F0 in some previous studies (e.g. Song et al., 2016; Walker et al., 2009), making it possible for animals to base their behavioural choices on the lower spectral edge of the sounds, rather than the sound's overall F0. Finally, most animal studies have not directly compared performance across human and non-human species (Bendor et al., 2012; Osmanski et al., 2013; Song et al., 2016), or have compared them across considerably different behavioural tasks (e.g. Shofner and Campbell, 2012 versus Shofner and Chaney, 2013), so differences in task demands might account for any apparent species differences. For example, the pitch difference thresholds of ferrets can differ by orders of magnitude between a go/no-go and 2AFC task (Walker et al., 2011).

The present study overcomes these limitations by directly comparing the pitch cues used by humans and ferrets on a common 2AFC pitch classification task. We first use a model of cochlear filtering to simulate the representation of periodic sounds in the inner ear in order to visualize the

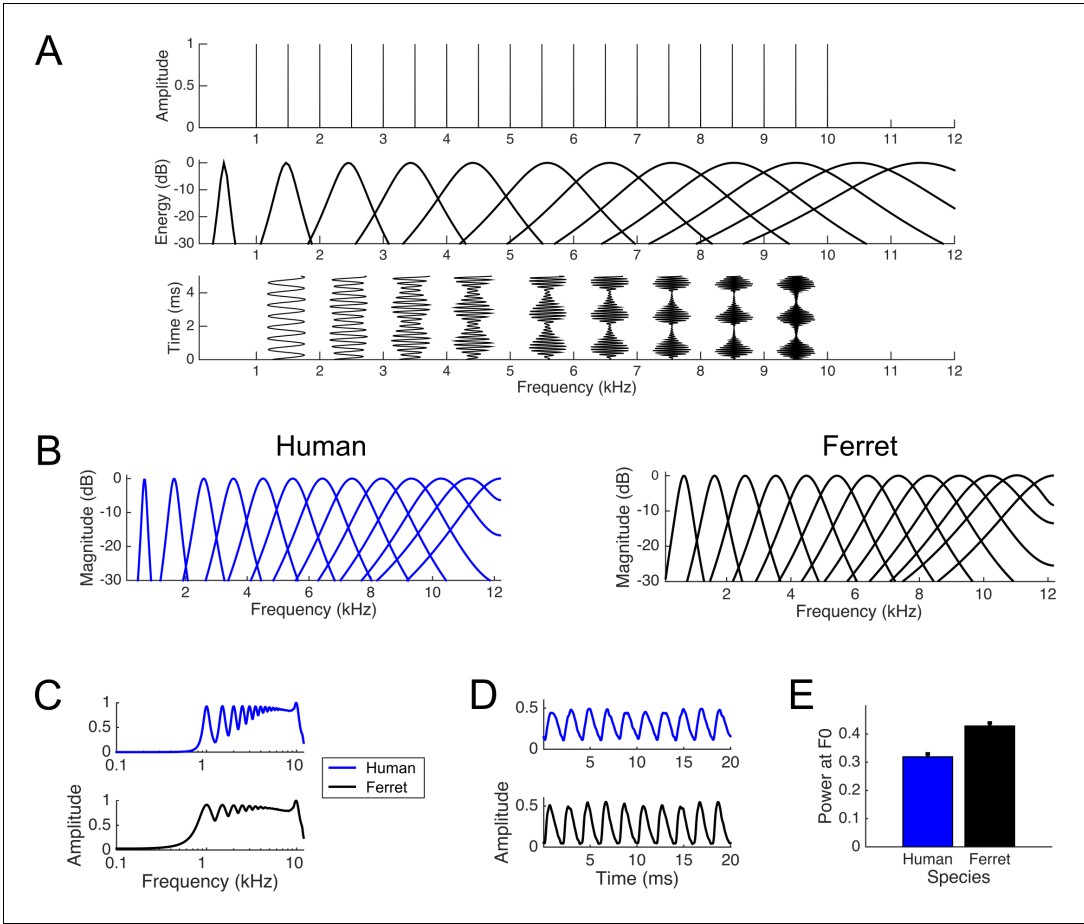

**Figure 1.** Simulated cochlear filters and their responses to a 500 Hz harmonic complex tone filtered from 1 to 10 kHz. (A) Illustration of the role of unresolved and resolved harmonics in periodicity encoding. Upper plot: Amplitude spectrum for the tone complex contains all harmonics from 1 to 10 kHz. This sound will evoke a pitch corresponding to 500 Hz. Middle plot: Cartoon of the cochlear filters centred on every second harmonic of 500 Hz, based on data from *Glasberg and Moore (1990)*. This illustrates that lower harmonics are resolved, while the cochlear filters corresponding to higher order harmonics respond to multiple harmonic components in the tone. Lower plot: The output of each of these cochlear filters is plotted throughout 5 ms of the tone complex. The resolved harmonics phase lock to the frequency of one harmonic, while unresolved harmonics beat at the sum of multiple harmonic components (i.e. 500 Hz), providing an explicit temporal representation of F0. B-E describe a computational model of the cochlear filter bank used to simulate representations of complex sounds in the ferret and human auditory nerve. Data are colour-coded for the human (blue) and ferret (black). (B) The frequency tuning of 15 example auditory nerve fibres is shown for the simulated human (left) and ferret (right) cochlea. C-E show analyses of the responses of human and ferret cochlear filter banks to the 500 Hz tone complex. (C) The response strengths of each of 500 auditory nerve fibres to the filtered tone complex were averaged across the duration of the sound, and plotted across the full range of centre frequencies. Many harmonics produce clearly resolvable activation peaks across fibres in the human cochlea (upper blue plot), but fewer harmonics are resolved in the ferret cochlea (lower black plot). (D) The temporal profile of the output of one simulated auditory nerve fibre with a centre frequency of 5 kHz is shown for the human (upper blue plot) and ferret (lower black plot) cochlea. (E) The power at 500 Hz in the output of each frequency filter, averaged across the full duration of the tone complex, is shown for the human (blue) and ferret (black) auditory nerve. For each species, these values were normalized by the maximal F0 power across all channels. The plot shows the mean (+standard error) normalized power at F0 across all auditory nerve fibres.

DOI: https://doi.org/10.7554/eLife.41626.002

effects of cochlear filter bandwidths on harmonic encoding. The simulations generated predictions about the availability of periodicity cues in the auditory nerve of each species. We then tested these predictions by training ferrets and humans to classify the pitch of a harmonic complex tone. We find differences in their dependence on resolved and unresolved harmonics, which in principle could be explained by differences in cochlear tuning between ferrets and humans.

## Results

### Simulating the filtering of complex tones in the ferret and human cochlea

Humans (*Glasberg and Moore, 1990*; *Shera et al., 2002*) are believed to have narrower cochlear filter bandwidths than ferrets (*Alves-Pinto et al., 2016*; *Sumner et al., 2018*; *Sumner and Palmer, 2012*) and other non-human animals (*Joris et al., 2011*; *Osmanski et al., 2013*; *Pickles and Comis, 1976*), and these physiological constraints may predispose them to rely on different acoustical cues to classify the pitch of complex tones. Specifically, sharper frequency tuning in humans should result in more resolvable harmonics across the human tonotopic map. On the other hand, if the bandwidth of an auditory nerve fibre is broader, its firing should phase lock more strongly to the beating of adjacent harmonics, potentially providing a stronger explicit representation of the temporal periodicity of F0 in ferrets than in humans.

To quantify and visualize these predicted effects, we modified a standard model of the cochlear filter bank (*Patterson et al., 1992*) to simulate the representation of tones along the human and ferret basilar membrane. The model followed existing literature (*Karajalainen, 1996*; *Patterson et al., 1992*; *Roman et al., 2003*), with parameters derived from either human psychophysics (*Glasberg and Moore, 1990*) or ferret auditory nerve recordings (*Sumner and Palmer, 2012*).

We chose human bandwidths estimated from simultaneous masking experiments (*Glasberg and Moore, 1990*). Because they incorporate effects of suppression, these bandwidths are likely to more effectively replicate the distribution of nerve excitation evoked by a complex tone than would bandwidths that omit these effects (e.g. those estimated with forward masking). It would be ideal to have comparable psychophysical measurements in ferrets, but the available measurements are too sparse to infer the dependence on frequency with confidence (*Sumner et al., 2018*). We instead used bandwidths from ferret auditory nerve tuning curves. These likely overestimate the degree of resolved harmonics, and thus provide a conservative estimate of the potential species difference.

As shown in *Figure 1B*, the cochlear filters are wider for the ferret auditory nerve than the human, even with the difference in the methods used to obtain the two sets of bandwidth estimates. In *Figure 1C–E*, we compare the human and ferret simulated responses to a 500 Hz missing F0 tone complex that we used as a training sound in our ferret behavioural experiment (described below).

When the instantaneous power of the cochlear filters is summed across the duration of the sound and plotted as a function of centre frequency, the individual low-numbered harmonics of the tone are more clearly resolved in the human cochlea than in the ferret (*Figure 1C*). This takes the form of deeper troughs in the activation of nerve fibres whose centre frequencies lie between the harmonic components of the sound. To visualize the temporal representation of the higher harmonics in the same stimulus, we plotted the output of a single nerve fibre (here, a fibre with a centre frequency of 5 kHz) throughout time (*Figure 1D*). In this case, the representation of the 500 Hz F0 is clearer in the ferret – the human cochlea produces weaker temporal modulation because fewer harmonics fall within the fibre's bandwidth.

We also examined whether the temporal representation of F0 was enhanced in the ferret cochlea across the full range of frequency filters. A Fourier transform was performed on the output of each fibre throughout a 200 ms steady-state portion of the sound. The power of the response at F0 was then expressed as a proportion of the overall power for that fibre. The results of this metric averaged across all fibres in the model are shown in *Figure 1E*. The average temporal representation of F0 was enhanced in the ferret compared to the human (Wilcoxon rank sum test; $z = 8.286$, $p=1.175\times10^{-16}$). In fact, this F0 representation metric was higher in the ferret than the human cochlear model across every pair of individually simulated auditory nerve fibres.

These simulations suggest that the ferret cochlea provides an enhanced representation of the envelope periodicity of a complex tone, as conveyed by spikes that are phase-locked to the F0 in

the auditory nerve. On the other hand, the human auditory nerve provides a better resolved representation of individual harmonics across the tonotopic array. It might thus be expected that these two types of cues would be utilized to different extents by the two species.

## Behavioural measures of pitch cue use in ferrets

To test the role of different pitch cues in ferret pitch perception, we trained five animals on a two-alternative forced choice (2AFC) task that requires 'low' and 'high' pitch judgements analogous to those used in human psychophysical tasks (*Figure 2A,B*). On each trial, a harmonic complex tone was presented at one of two possible fundamental frequencies. Ferrets were given water rewards for responding at the right nose-poke port for a high F0, and at the left port for a low F0. Incorrect responses resulted in a time-out. We began by training four ferrets to classify harmonic complex tones with an F0 of 500 and 1000 Hz, with a repeating pure tone presented at 707 Hz (the midpoint on a logarithmic scale) for reference before each trial. Two of these animals, along with one naïve ferret, were then trained on the same task using target F0 values of 150 and 450 Hz and a 260 Hz pure tone reference. In both cases, the integer ratios between the F0s to be discriminated allowed us to match their spectral bandwidths exactly, so that ferrets could not solve the task based on the frequency range of the sound (*Figure 3*; first four rows). Rather, the animals had to discriminate sounds based on some cue to the F0. After completing several pre-training stages to habituate the animals to the apparatus and sound presentation (see Materials and methods), the ferrets learned to perform the pitch classification task within 22 ± 3 (mean ± standard deviation) days of twice daily training.

Once the ferrets learned to perform this simple 2AFC task, we incorporated 'probe trials' into the task in order to determine which acoustical cues they were using to categorize the trained target sounds. Probe trials made up 20% of trials in a given session, and were randomly interleaved with the 'standard' trials described above. On probe trials, an untrained target stimulus was presented following the pure tone reference, and the ferret received a water reward regardless of its behavioural choice. This task design discouraged ferrets from learning to use a different strategy to classify the probe sounds.

The inner ear is known to produce distortion in response to harmonic tones that can introduce energy at the fundamental frequency to the basilar membrane response, even for missing-fundamental sounds (*Robles et al., 1991*). These distortion products could in principle counter our attempts to match the spectral bandwidths of the sounds, since they could cause the lowest frequency present in the ear to differ as a function of F0. To determine if the ferrets relied on such cochlear distortion products to classify tones in our task, we added pink noise to the stimulus on 20% of randomly interleaved probe trials at an intensity that is known to be more than sufficient to mask cochlear distortion products in humans (*Norman-Haignere and McDermott, 2016*; *Pressnitzer and Patterson,*

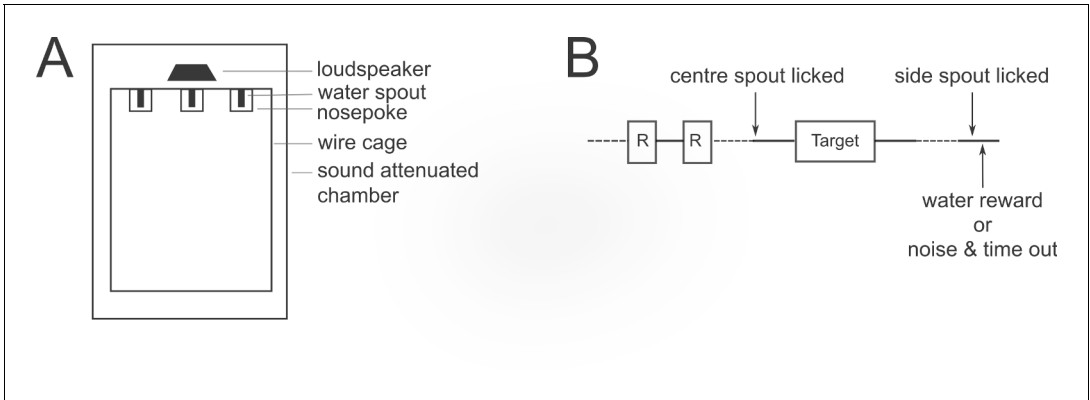

**Figure 2.** Psychophysical task design. (**A**) Schematic of the ferret testing apparatus, viewed from above. (**B**) Schematic of one trial in the 2-alternative forced choice pitch classification task. The target tone complex could be lower or higher in F0 than the reference pure tone (**R**). Dotted lines indicate time durations that are variable, depending on the animal's behaviour.
DOI: https://doi.org/10.7554/eLife.41626.003

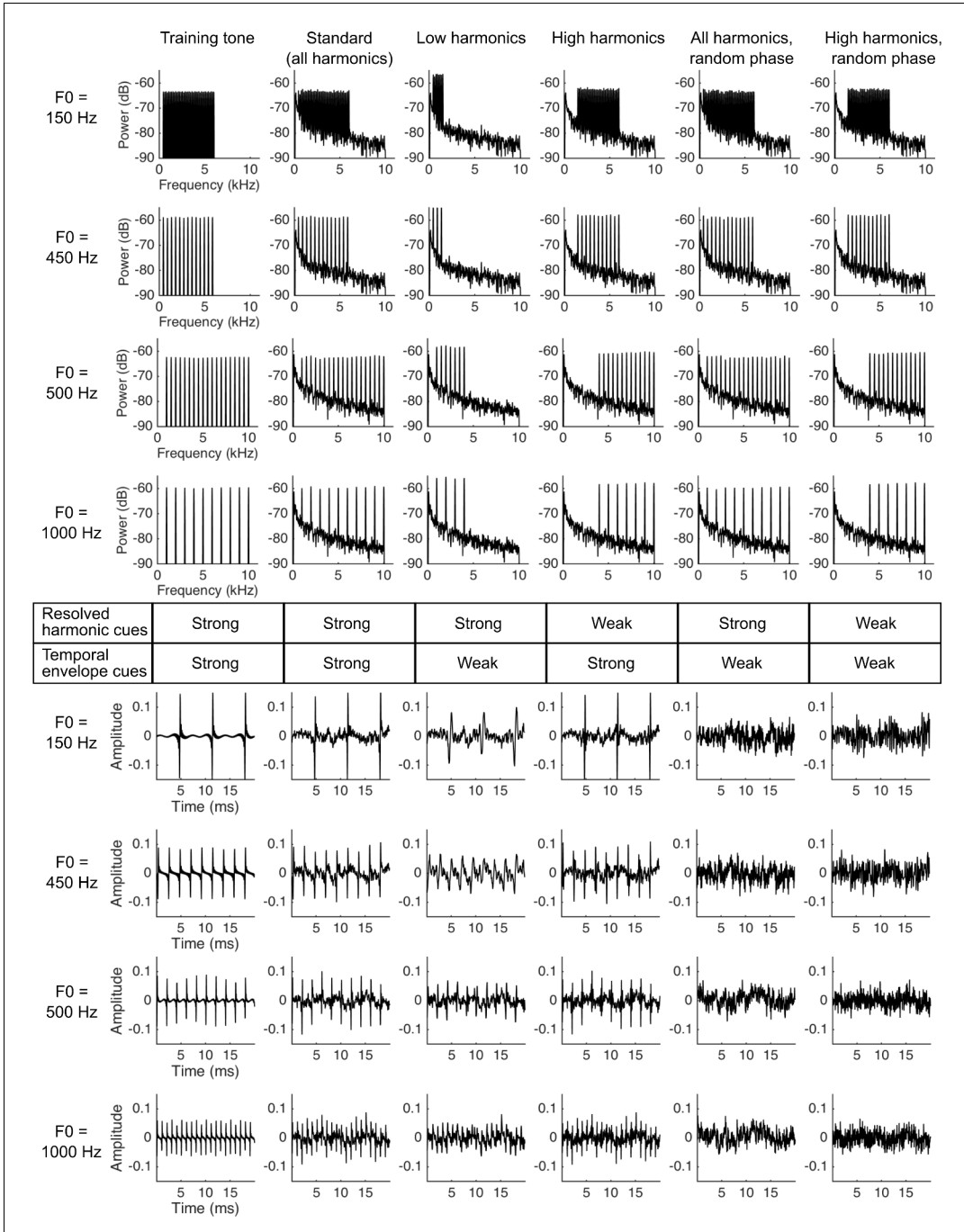

**Figure 3.** Stimuli used in the ferret pitch classification task. Plots show the training tone (left column), standard stimulus (second column) and four probe stimuli (columns 3–6) used in the psychophysical task. There were two F0 ranges, with target stimuli with F0s of either 150 and 450 Hz, or 500 and 1000 Hz, indicated to the left of each row of plots. The top four rows show the power spectra of each target sound, while the bottom four rows plot a 20 ms excerpt of the corresponding sound waveform. The table in the middle of the figure indicates whether resolved harmonics (row 1) or temporal envelope (row 2) F0 cues are strong or weak in each stimulus. A pure tone reference sound was used in all experimental stages.
DOI: https://doi.org/10.7554/eLife.41626.004

*2001*). Ferrets performed more poorly on probe trials than on standard trials (paired t-test; t = 4.346, p=0.005), as expected for an auditory discrimination task performed in noise. However, they performed the pitch classification at 71.85 ± 9.60% correct (mean ± standard deviation) with the

noise masker, which is well above chance (1-sample t-test; t = 6.025, p=0.001). This suggests that ferrets did not rely on cochlear distortion products to solve our task.

We next moved to the main testing stage of our behavioural experiment, which aimed to determine if ferrets use resolved harmonics, temporal envelope periodicity, or both of these cues to identify the F0 of tones. All tone complexes, both the standard and probe stimuli, were superimposed on a pink noise masker. Our auditory nerve model (above) allowed us to estimate which harmonics in the tone complexes would be resolved in the ferret auditory nerve (*Figure 4A*) (*Moore and Ohgushi, 1993*). This analysis suggests that our standard tones contained both resolved and unresolved harmonics for ferret listeners, as intended. We constructed four types of probe stimuli based on our resolvability estimates: (1) 'Low Harmonic' tones containing only harmonics that we expected

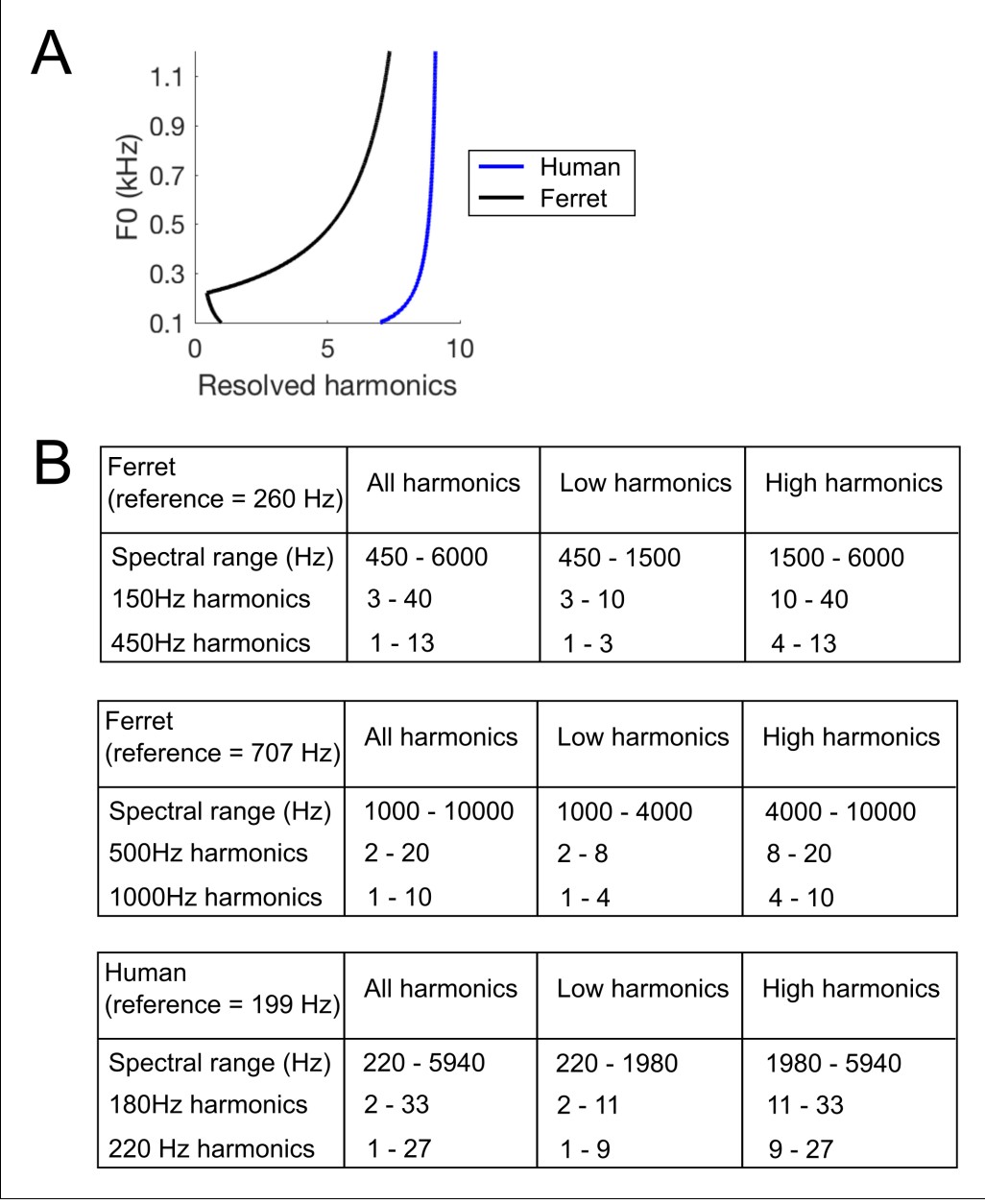

**Figure 4.** Harmonic content of stimuli. (**A**) The number of resolved harmonics was estimated over a range of F0s, for the human (blue) and ferret (black) cochlea. (**B**) The frequency ranges and numbers of harmonic partials included in each stimulus. As per convention, F0 is deemed to be harmonic 1.
DOI: https://doi.org/10.7554/eLife.41626.005

to be resolved; (2) 'High Harmonic' tones containing harmonics presumed to be less well resolved; (3) 'All Harmonics Random Phase' probes containing the full set of harmonics present in the standard tone, but whose phases were independently randomized in order to flatten the temporal envelope; and (4) 'High Harmonics Random Phase' stimuli with the same randomization of harmonic phases, but containing only presumptively unresolved harmonics. It is possible that the highest resolved harmonic for the ferret might have been overestimated by our procedure, given that cochlear filter bandwidths were taken from auditory nerve tuning curves for pure tones and thus did not incorporate effects of lateral suppression. If so, fewer harmonics might actually have been resolved by ferrets in the 'Low Harmonic' stimulus than indicated by our simulations. However, this would not change the rationale for using these stimuli or the interpretation of the behavioral data.

The spectral ranges of these stimuli are given in *Figure 4B*, and the spectra and audio waveforms (showing the temporal envelope periodicity) are illustrated in *Figure 3A*. Ferrets were again given water rewards irrespective of their behavioural choice on probe trials, in order to avoid reinforcing different pitch classification strategies across probe stimuli.

The performance of ferrets on the standard and probe stimuli is shown in *Figure 5A*. Data for individual subjects are shown in *Figure 5—figure supplement 1A*. A repeated-measures 3-way ANOVA was carried out on the three ferrets trained with both references. This analysis indicated that performance varied with stimulus type (i.e., the standard and four probe stimuli) ($F_{(4,8)}$ = 10.540, p=0.003), but not across subjects ($F_{(2,8)}$ = 1.060; p=0.391) or the two reference conditions (i.e., 260 and 707 Hz) ($F_{(1,8)}$ = 0.438, p=0.576). Scores did not significantly vary across individual ferrets in either the 260 Hz (2-way ANOVA; $F_{(2,8)}$ = 0.366, p=0.704) or 707 Hz condition (2-way ANOVA; $F_{(3,12)}$ = 2.063, p=0.158), so data collected from the same animals in these two conditions were treated as independent measurements in subsequent analyses.

To assess the acoustical cues used by animals to solve the pitch classification task, we compared ferrets' performance on the standard trials with that on each of the four probe trial types (repeated measures 2-way ANOVA, Tukey's HSD test). Compared to their performance on standard trials, ferrets showed impaired performance on probes that contained only low harmonics (p=0.001), but performed well when only high harmonics were presented (p=1.000). Their performance was impaired when we randomized the phases of the high harmonics (p=0.002). Phase randomization also impaired performance when the full set of harmonics (both resolved and unresolved; 'All Harmonics Random Phase') were present (p=$2.173\times10^{-5}$). This pattern of results suggests that ferrets rely more strongly on the temporal envelope periodicity (produced by unresolved harmonics) than on resolved harmonics to classify the pitch of harmonic tone complexes, unlike what would be expected for human listeners.

## Comparison of human and ferret pitch classification performance

Humans were trained on a similar pitch classification task to the one described for ferrets in order to best compare the use of pitch cues between these two species. Participants were presented with harmonic complex tones and classified them as high or low. A training phase was used to teach participants the high and low F0s.

We tested human listeners using the same types of standard and probe stimuli as in the final stage of ferret testing described above. As the pitch discrimination thresholds of human listeners are known to be superior to those of ferrets (*Walker et al., 2009*), we adapted the target F0s (180 and 220 Hz) and harmonic cut-offs for human hearing (*Figure 4*). The constraints of using a smaller F0 difference necessitated a different lowest frequency component between the two F0s for the 'All Harmonic' and 'Low Harmonic' conditions, unlike the ferret stimuli. However, the spectral edge was higher for the lower F0 (360 Hz vs. 220 Hz), and thus is unlikely to have provided a cue that could be used to correctly perform the task (particularly given that we did not provide feedback). Because the F0 difference differed between species, the between-species comparison of interest here is not the difference in absolute scores on the task, but the pattern of performance across probe conditions.

Human listeners also showed varied pitch classification performance across the standard and probe stimuli (repeated-measures 2-way ANOVA; $F_{(4,60)}$ = 36.999, p=$1.443\times10^{-15}$). However, a different pattern of performance across stimuli was observed for human subjects than for ferrets (*Figure 5B*). Tukey's HSD tests indicated that human listeners were significantly impaired when resolved harmonics were removed from the sounds, as demonstrated by impairments in the 'High Harmonic' probes with (p=$9.922\times10^{-9}$) and without (p=$1.029\times10^{-8}$) randomized phases.

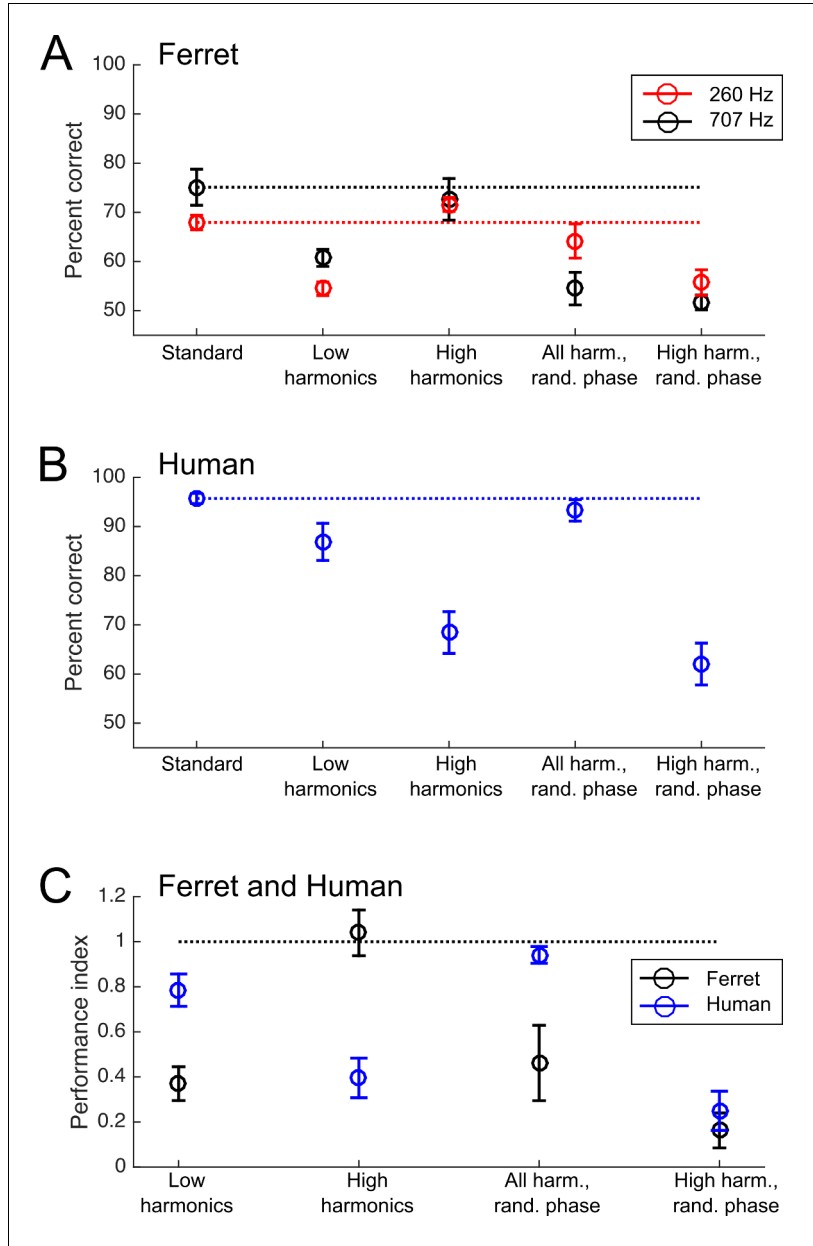

**Figure 5.** Pitch classification performance of ferrets and humans. (**A**) Ferrets' percent correct scores on the pitch classification task are plotted for the standard tone trials (left) and each of the four probe stimuli (right). The results of testing with the 260 Hz reference (150 and 450 Hz targets; red) and 707 Hz reference (500 and 1000 Hz targets; black) are plotted separately. (**B**) Humans' pitch classification performance is plotted, as in (**A**). (**C**) Performance for each of the four probe stimuli is expressed as the ratio of the percentage correct score and that achieved with the standard training tone stimulus. Data are shown for ferrets (black) and humans (blue). Values of 0 indicate that subjects performed at chance for the probe stimulus, while one indicates that they classified the F0 of the probe as accurately as the standard stimulus. Error bars shown mean ± standard error. Individual data for (**A**) and (**B**) are shown in *Figure 5—figure supplement 1*.

DOI: https://doi.org/10.7554/eLife.41626.006

The following source data and figure supplement are available for figure 5:

**Source data 1.** Stimuli and data from psychophysical experiments.
DOI: https://doi.org/10.7554/eLife.41626.008
**Figure supplement 1.** Pitch classification performance of individual ferrets and humans, as shown in *Figure 5A and B*.
DOI: https://doi.org/10.7554/eLife.41626.007

Conversely, no impairment was observed when resolved harmonics were available, regardless of whether the phases of stimuli were randomized ('All Harmonics Random Phase' condition; p=0.959) or not ('Low Harmonics' condition; p=0.101). Data for individual subjects are shown in *Figure 5—figure supplement 1B*. These results are all consistent with the wealth of prior work on human pitch perception, but replicate previously reported effects in a task analogous to that used in ferrets.

The performance for each probe type, relative to performance on the standard stimuli, is directly compared between the two species in *Figure 5C*. Here, a score of 1 indicates that the subject performed equally well for the standard tone and the probe condition, while a score of 0 indicates that the probe condition fully impaired their performance (reducing it to chance levels). This comparison illustrates the differences in acoustical cues underlying ferret and human pitch classifications. Consistent with the relative prominence of the cues in our model simulations, we found that while ferrets were impaired only when temporal envelope cues from unresolved harmonics were disrupted, humans continued to classify the target pitch well in the absence of temporal envelope cues, so long as resolved harmonics were present. This was confirmed statistically as a significant interaction between species and stimulus type on performance (2-way ANOVA; $F(4,135) = 14.720$, $p=5.274\times10^{-10}$). The two species thus appear to predominantly rely on distinct cues to pitch.

## Discussion

We used a combination of cochlear modelling and behavioural experiments to examine the use of pitch cues in ferrets and human listeners. Our model simulations illustrated how broader cochlear filter widths in ferrets result in fewer resolved harmonics and a more enhanced representation of temporal envelopes than the human cochlea, as suggested by previous authors (*Cedolin and Delgutte, 2010*; *Shofner and Chaney, 2013*). Based on this result, we predicted that the pitch judgments of ferrets would rely more strongly on temporal envelope cues than that of human listeners. Our behavioural experiments directly compared the use of pitch cues in the two species and found that this is indeed the case. Our results provide the first unambiguous dissociation of pitch mechanisms across species, by utilizing the same task across species, and provide an illustration of the potential consequences of species differences in cochlear tuning.

### Findings in other species

Human listeners have consistently been found to have better pitch discrimination thresholds when stimuli contain resolved harmonics (*Bernstein and Oxenham, 2003*; *Kaernbach and Bering, 2001*; *Moore et al., 1985*; *Ritsma, 1967*; *Shackleton and Carlyon, 1994*). Moreover, cortical responses to pitched sound in humans are stronger for resolved than unresolved harmonics, mirroring perceptual sensitivity (*Norman-Haignere et al., 2013*; *Penagos et al., 2004*). The results of our human experiments are thus fully consistent with this large body of prior work, while enabling comparison with non-human animals. Because most natural sounds contain both low- and high-numbered harmonics, humans may learn to derive pitch primarily from resolved harmonics even when temporal envelope cues are also available, and are thus less equipped to derive pitch from unresolved harmonics alone. This would explain the drop in performance when resolved harmonic cues were removed on probe trials in our experiment.

Our cochlear simulations suggest that harmonic resolvability is worse for ferrets than human listeners, as a consequence of their previously demonstrated differences in cochlear filter widths (*Sumner et al., 2018*). As a result, they may conversely learn to rely more on temporal pitch cues when estimating pitch from natural sounds, leading to poorer performance for low harmonic tone complexes. In fact, these species differences may be even more pronounced than suggested by our simulations, as ferret filters appear to exhibit broader bandwidths when measured with the simultaneous masking paradigms from which the human bandwidths in our simulations were obtained (*Sumner et al., 2018*). Similarly, human cochlear bandwidths have been shown to be sharper when estimated from otoacoustic emissions or forward masking paradigms (*Shera et al., 2002*) than from a simultaneous masking task (*Glasberg and Moore, 1990*).

Many non-human mammals are thought to have wider cochlear bandwidths than humans (*Joris et al., 2011*; *Liberman, 1978*; *Sumner et al., 2018*; *Temchin et al., 2008*), and so we might expect temporal cues to dominate their pitch decisions as we have observed in ferrets. The few studies to directly address F0 cue use in pitch judgments by non-human animals have raised the

possibility of species differences in pitch perception, but have relied on go/no-go tasks that differ from standard psychophysical tasks used in humans. For instance, studies in gerbils suggest that they primarily use temporal cues to detect an inharmonic component in a tone complex (*Klinge and Klump, 2010*; *Klinge and Klump, 2009*). Chinchillas were similarly shown to detect the onset of a periodic sound following a non-periodic sound using temporal, rather than resolved harmonic, cues (*Shofner, 2002*). While these studies did not explicitly compare the use of resolved and unresolved pitch cues, their findings are consistent with ours regarding the importance of temporal cues in non-human species, and the authors similarly proposed that species differences could be explained by differences in peripheral tuning.

Marmosets, on the other hand, appear to be influenced by harmonic phases when detecting changes in F0 only when resolved harmonics are omitted from the stimulus (*Bendor et al., 2012*; *Osmanski et al., 2013*; *Song et al., 2016*). This suggests that temporal cues are only salient for this species when they occur in unresolved harmonics. Similarly to humans, marmosets were found to detect smaller changes in F0 when harmonics were resolved than when only unresolved harmonics were available (*Song et al., 2016*). Comparable studies have yet to be carried out in other non-human primates, so it remains unclear whether primates are special in the animal kingdom in their dependence on resolved harmonic cues. We note also that the behavioural task used in previous marmoset experiments (*Bendor et al., 2012*; *Osmanski et al., 2013*; *Song et al., 2016*) required animals to detect a change in F0, whereas the task employed in this study required ferrets to label the direction of F0 changes. Ferrets show an order of magnitude difference in pitch acuity on these two tasks (*Walker et al., 2011*), raising the possibility that primates might as well.

The use of probe trials without feedback in the present experiment allowed us to determine which acoustical cues most strongly influenced listeners' pitch judgements. The ferrets relied predominantly on temporal cues under these conditions, but our results do not preclude the possibility that they could also make pitch judgments based on resolved harmonics if trained to do so. Indeed, although human listeners rely on resolved harmonics under normal listening conditions, we can also extract pitch from unresolved harmonics when they are isolated (*Moore et al., 1985*; *Ritsma, 1967*; *Shackleton and Carlyon, 1994*). Our simulations show that multiple harmonics could potentially be resolved on the ferret cochlea, depending on the F0 (*Figure 4A*). Consequently, if specifically trained to do so, one might expect ferrets to be able to derive F0 from these harmonics using the same template matching mechanisms proposed for human listeners (*Goldstein, 1973*; *Shamma and Klein, 2000*). As discussed above, it is also important to note that the relationship between harmonic resolvability and auditory nerve tuning is not fully understood, and nonlinearities in response to multiple frequency components could cause resolvability to be worse than that inferred from isolated auditory nerve fibre measurements.

Overall, the available evidence fits with the idea that pitch judgments are adapted to the acoustical cues that are available and robust in a particular species, with differences in cochlear tuning thus producing cross-species diversity in pitch perception. A similar principle may be at work within human hearing, where listeners appear to rely on harmonicity for some pitch tasks and spectral changes in others, potentially because of task-dependent differences in the utility of particular cues (*McPherson and McDermott, 2018*). The application of normative models of pitch perception will likely provide further insight into the relative importance of these cues.

## Implications for neurophysiological work

A better understanding of the similarities and differences in pitch processing across species is essential for relating the results of neurophysiological studies in animals to human pitch perception. The present experiments suggest that ferrets, a common animal model in studies of hearing (e.g. *Atilgan et al., 2018*; *Bizley et al., 2013*; *Fritz et al., 2003*; *Schwartz and David, 2018*), can estimate F0 from the temporal envelopes of harmonic complex tones. Our data indicate that ferrets generalize across sounds with different spectral properties (including wideband sounds, sounds in noise, and sounds containing only high harmonics) without relying on explicit energy at the F0. In this respect, ferrets appear to have a pitch percept, even though the cues underlying it are apparently weighted differently than in human pitch perception.

It is interesting that ferrets appear to favour temporal cues over resolved harmonic cues even for F0s as high as 1000 Hz. While this periodicity is well below the phase locking limit of the ferret auditory nerve (*Sumner and Palmer, 2012*), it falls above a reported temporal pitch limit in cats

(*Chung and Colavita, 1976*) and humans (*Carlyon and Deeks, 2002*; *Macherey and Carlyon, 2014*). We thus cannot rule out the possibility that ferrets were actually performing the 707 Hz reference task by detecting the absence of temporal modulation at 1000 Hz. However, the similar pattern of performance for the lower pair of reference F0s (150 and 450 Hz) is less plausibly explained in this way, and supports the idea that the ferrets were basing their judgments on temporal pitch cues. It is also possible that the temporal modulation limit of pitch is higher in an animal that relies more on this cue.

The existing literature might be taken to suggest that primates are more appropriate animal models for examining the role of resolved harmonics in human pitch perception, as they appear to be more like humans in their use of this cue (*Bendor et al., 2012*; *Osmanski et al., 2013*; *Song et al., 2016*). On the other hand, our data suggest that ferrets are a powerful animal model for evaluating temporal models of pitch extraction (e.g. *Meddis et al., 1997*). Like cochlear implant users, ferrets have broad cochlear frequency filters that may limit their use of resolved harmonic cues.

## Materials and methods

### Experimental subjects

#### Ferrets (Mustela putorius furo)

Five adult female pigmented ferrets (aged 6–24 months) were trained in this study. Power calculations estimated that five animals was the minimum appropriate sample size for 1-tailed paired comparisons with alpha = 5%, a medium (0.5) effect size, and beta = 20%. Ferrets were housed in groups of 2–3, with free access to food pellets. Training typically occurred in runs of 5 consecutive days, followed by two days rest. Ferrets could drink water freely from bottles in their home boxes on rest days. On training days, drinking water was received as positive reinforcement on the task, and was supplemented as wet food in the evening to ensure that each ferret received at least 60 ml/kg of water daily. Regular otoscopic and typanometry examinations were carried out to ensure that the animals' ears were clean and healthy, and veterinary checks upon arrival and yearly thereafter confirmed that animals were healthy. The animal procedures were approved by the University of Oxford Committee on Animal Care and Ethical Review and were carried out under license from the UK Home Office, in accordance with the Animals (Scientific Procedures) Act 1986.

#### Humans

The pitch classification performance of 16 adult humans (nine male, ages 18–53 years; mean age = 25.3 years) was also examined, which provided a 60% beta in the power calculations described for ferrets. All subjects reported having normal hearing. All experimental procedures on humans were approved by the Committee on the Use of Humans as Experimental Subjects at MIT.

### Method details

#### Cochlear filter simulations

We used a cochlear filter bank previously developed by *Patterson et al. (1992)* and implemented by *Slaney (1993)* to simulate representations of sounds on the basilar membrane. The model simulates the response of the basilar membrane to complex sounds as a set of parallel Gammatone filters, each with a different characteristic frequency and Equivalent Rectangular Bandwidth (ERB). In order to compare the representation of harmonic tone complexes in the human and ferret cochlea, we modified this model to use filter constants derived from either psychophysical estimates of human cochlear filters (*Glasberg and Moore, 1990*), or ferret auditory nerve recordings (*Sumner and Palmer, 2012*). Based on these sources, the equivalent rectangular bandwidth of filter $i$ in the human cochlea was calculated as:

$$\mathrm{ERB}_i = \mathrm{f}_i / (12.7 * (\mathrm{f}_i / 1000)0.3),$$

where $\mathrm{f}_i$ is the centre frequency of the filter in Hz.

For the ferret cochlea, the equivalent rectangular bandwidth of each filter was estimated using the following linear fit to the data in *Sumner and Palmer (2012)*:

$$\mathrm{ERB}_i = \mathrm{f}_i/8.9047 + 209.6149.$$

The output of each channel in the above Gammatone filter bank was half-wave rectified and then compressed (to the power of 0.3) to simulate transduction of sound by inner hair cells. Finally, the output was low-pass filtered at 3 kHz (FIR filter, passband 3 kHz, stopband 4 kHz, attenuation 60 dB) to reflect the phase locking limit of auditory nerve fibres. This model architecture is similar to that used in previous studies (e.g. *Karajalainen, 1996*; *Roman et al., 2003*).

## Training apparatus

Ferrets were trained to discriminate sounds in custom-built testing chambers, constructed from a wire mesh cage (44 × 56 × 49 cm) with a solid plastic floor, placed inside a sound-insulated box lined with acoustic foam to attenuate echoes. Three plastic nose poke tubes containing an inner water spout were mounted along one wall of the cage: a central 'start spout' and two 'response spouts' to the left and right (*Figure 2A*). Ferrets' nose pokes were detected by breaking an infrared LED beam across the opening of the tube, and water was delivered from the spouts using solenoids. Sound stimuli, including acoustic feedback signals, were presented via a loudspeaker (FRS 8; Visaton, Crewe, UK) mounted above the central spout, which had a flat response (±2 dB) from 0.2 to 20 kHz. The behavioural task, data acquisition, and stimulus generation were all automated using a laptop computer running custom MATLAB (The Mathworks, Natick, MA, USA) code, and a real-time processor (RP2; Tucker-Davis Technologies, Alachua, FL, USA).

## Pre-training

Ferrets ran two training sessions daily, and typically completed 94 ± 24 trials per session (mean ± standard deviation). Several pre-training stages were carried out to shape animals' behaviour for our classification task. In the first session, animals received a water reward whenever they nose poked at any of the spouts. Next, they received water rewards only when they alternated between the central and peripheral spouts. The water reward presented from the peripheral response spouts (0.3–0.5 ml per trial) was larger than that presented at the central start spout (0.1–0.2 ml per trial). The animal was required to remain in the central nose poke for 300 ms to receive a water reward from that spout.

Once animals performed this task efficiently, sound stimuli were introduced in the next session. At the start of each trial, a repeating pure tone 'reference' (200 ms duration, 200 ms inter-tone interval, 60 dB SPL) was presented to indicate that the central spout could be activated. Nose poking at the central spout resulted in the presentation of a repeating complex tone 'target' (200 ms duration, 200 ms inter-tone interval, 70 dB SPL) after a 100 ms delay. The animal was again required to remain at the centre for 300 ms, and early releases now resulted in the presentation of an 'error' broadband noise burst (200 ms duration, and 60 dB SPL) and a 3 s timeout before a new trial began. The target tone could take one of two possible F0 values, which corresponded to rewards at one of the two peripheral spouts (right rewards for high F0 targets, and left for low F0s). For all training and testing stages, the target tones contained harmonics within the same frequency range, so that animals could not use spectral cut-offs to classify the sounds. The target tone continued to play until the animal responded at the correct peripheral spout, resulting in a water reward. Once the animals could perform this final pretraining task with >70% accuracy across trials, they advanced to pitch classification testing.

## Testing stages and stimuli

The complex tone target was presented only once per trial, and incorrect peripheral spout choices resulted in an error noise and a 10 s timeout (*Figure 2B*). After such an error, the following trial was an error correction trial, in which the F0 presented was the same as that of the previous trial. These trials were included to discourage ferrets from always responding at the same peripheral spout. If the ferret failed to respond at either peripheral spout for 14 s after target presentation, the trial was restarted.

The reference pure tone's frequency was set to halfway between the low and high target F0s on a log scale. We examined ferrets' pitch classification performance using two pairs of complex tone targets in separate experimental blocks: the first with F0s of 500 and 1000 Hz (707 Hz reference),

and the second with 150 and 450 Hz targets (260 Hz reference). The 150 and 450 Hz targets were chosen to overlap with the F0 range that we tested in human listeners (below). The 500 and 1000 Hz condition was included as ferrets often perform better on pitch discrimination tasks in this range than on sounds with lower F0s (*Walker et al., 2009*). Four ferrets were trained on the 707 Hz reference. Two of these animals, plus an additional naive animal, were trained on the 260 Hz reference. In each case, testing took place over three stages, in which the ferret's task remained the same but a unique set of stimulus parameters was changed (*Figures 3* and *4*), as outlined below. Ferrets were allocated to the 260 and 707 Hz reference conditions based on their availability at the time of testing.

*Stage 1:* Target sounds were tone complexes, containing all harmonics within a broad frequency range (specified in *Figure 4B*). The pairs of target stimuli were chosen to be either one octave (a factor of two; 500 and 1000 Hz) or a factor of three (150 and 450 Hz) apart so that their ranges of harmonics could be matched exactly in spectral range. When an animal performed this task >75% correct on three consecutive sessions, ($32.8 \pm 7.1$ sessions from the beginning of training; mean ± standard deviation; n = 4 ferrets), they moved to Stage 2.

*Stage 2:* On 80% of trials, the same standard target tones from Stage one were presented. The other 20% of trials were 'probe trials', in which the ferret was rewarded irrespective of the peripheral spout it chose, without a timeout or error correction trial. Probe trials were randomly interleaved with standard trials. The probe stimuli differed only by the addition of pink noise (0.1–10 kHz) to the target sounds, in order to mask possible cochlear distortion products at F0. The level of the noise masker was set so that the power at the output of a Gammatone filter centred at the F0 (with bandwidth matched to ferret auditory nerve measurements in that range [*Sumner and Palmer, 2012*]) was 5 dB below the level of the pure tone components of the target. This is conservative because distortion products are expected to be at least 15 dB below the level of the stimulus components based on measurements in humans (*Norman-Haignere and McDermott, 2016*; *Pressnitzer and Patterson, 2001*). When an animal performed this task >75% correct on three consecutive sessions, they moved to stage 3.

*Stage 3:* The probe stimulus from Stage two served as the 'Standard' sound on 80% of trials, and all stimuli (both the standard and probes) included the pink noise masker described above. Twenty percent of trials were probe trials, as in Stage 2, but this stage contained tones manipulated to vary the available pitch cues. We estimated the resolvability of individual harmonics using ERB measurements available in previously published auditory nerve recordings (*Sumner and Palmer, 2012*). For a given F0, the number of resolved harmonics was approximated as the ratio of F0 and the bandwidth of auditory nerve fibres with a characteristic frequency at that F0, as described by *Moore and Ohgushi (1993)*, and applied by *Osmanski et al. (2013)*. This measure yielded between 1 and 8 resolved harmonics for ferrets, depending on the F0 (*Figure 4A*). Four types of probe stimuli were presented: (1) 'Low Harmonics', which contained only harmonics presumed to be resolved; (2) 'High Harmonics', comprised of harmonics presumed to be unresolved; (3) 'All Harmonics Random Phase', which contained the same set of harmonics as the standard, but whose phases were independently randomized in order to reduce temporal envelope cues for pitch; and (4) 'High Harmonics Random Phase', which contained the harmonics present in 'High Harmonics' stimuli, but with randomized phases. The bandpass cutoffs for the probe stimuli were chosen so that the 'Low Harmonic', but not 'High Harmonic', probes contained resolved harmonics for ferret listeners. Each probe stimulus was presented on at least 40 trials for each ferret, while the standard was tested on over 1000 trials per ferret.

## Human psychophysical task

Human subjects were tested on a pitch classification task that was designed to be as similar as possible to Stage 3 of ferrets' task (see above). 16 subjects discriminated target F0s of 180 and 220 Hz. Due to the smaller F0 difference required to make the task difficult enough to challenge human listeners (*Walker et al., 2009*), it was not possible to match the lower spectral edge of the 'Low Harmonic' and 'All Harmonic' stimuli as we did for ferrets. However, the stimuli were set such that the higher F0 target had the lower spectral edge. As a result, these edge cues were incongruent with the F0. Because feedback was not provided, it is unlikely that subjects could have learned to associate a lower spectral edge with the higher F0 and vice versa. This stimulus confound thus if anything

is likely to have made the task more difficult in the 'Low Harmonic' and 'All Harmonic' conditions. Because our main finding is that the relative performance of humans was better than that of ferrets in these conditions, it seems unlikely to have influenced the key results.

In the psychophysical task, human listeners were presented with the same classes of stimuli described above for ferrets. The frequency ranges included in the probe stimuli are listed in *Figure 4B*. Sounds were presented over headphones (Sennheiser HD280) in a sound attenuated booth (Industrial Acoustics, USA). A repeating reference pure tone (200 ms duration, 200 ms inter-tone interval, 60 dB SPL) was presented at the start of a trial, and the subject initiated the target harmonic tone complex (200 ms duration, 70 dB SPL) presentation with a keypress. Text on a computer monitor then asked the subject whether the sound heard was the low or high pitch, which the subjects answered via another keypress (1 = low, 0 = high). Feedback was given on the monitor after each trial to indicate whether or not the subject had responded correctly. Incorrect responses to the standard stimuli resulted in presentation of a broadband noise burst (200 ms duration, and 60 dB SPL) and a 3 s timeout before the start of the next trial. Error correction trials were not used for human subjects, as they did not have strong response biases. Standard harmonic complex tones were presented on 80% of trials, and the four probes ('Low Harmonics', 'High Harmonics', 'All Harmonics Random Phase', and 'High Harmonics Random Phase') were presented on 20% of randomly interleaved trials. Feedback for probe trials was always 'correct', irrespective of listeners' responses. Humans were given 10 practice trials with the standard stimuli before testing, so that they could learn which stimuli were low and high, and how to respond with the keyboard. Each probe stimulus was tested on 40 trials for each subject, while the standard was tested on 680 trials per subject.

## Quantification and statistical analysis

### Psychophysical data analysis

Error correction trials were excluded from all data analysis, as were data from any testing session in which the subject scored less than 60% correct on standard trials. T-tests and ANOVAs with an alpha of 5% were used throughout to assess statistical significance.

Because only 3 of the four ferrets were trained on both references (the final ferret was only trained on the 707 Hz reference condition), the repeated measures ANOVA used to analyse the ferret data was limited to these three ferrets. This ANOVA indicated that performance effects did not vary significantly across animals. We therefore performed the rest of our analysis while treating ferrets as independent measures in the two conditions, allowing us to include all four ferrets (otherwise the ANOVA would be unbalanced). Because animal behaviour is very labour intensive to collect, we decided to sacifice the repeated measure analysis to include the fourth ferret. In any case, our results were sufficiently robust so as to not require the additional sensitivity of a repeated measures analysis.

Error bars in *Figures 1* and *5* show mean ± standard errors. Further details of all statistical tests described here are provided as tables (*Supplementary files 1a-1k*).

Because humans produced higher percent correct scores overall than ferrets on the behavioural task, we normalized probe scores against the standard scores when directly comparing performance between species. The score of each species in each probe condition was represented as:

$$Pnorm_{ai} = (P_{ai} - 50)/(S_a - 50),$$

where $Pnorm$ is the normalized probe score for species $a$ on probe $i$, $P_{ai}$ is the percent correct score for species $a$ on probe $i$, and $S_a$ is the percent correct score of species $a$ on the standard trials. If the performance of species $a$ is unimpaired for a given probe stimulus $i$ relative to the standard stimulus, then $Pnorm_{ai}$ will equal 1. If the listeners are completely unable to discriminate the F0 of the probe, then $Pnorm_{ai} = 0$.

The data and custom software developed in this manuscript are available on the Dryad archive.

## Acknowledgments

This work was supported by a BBSRC New Investigator Award (BB/M010929/1) and a DPAG Early Career Fellowship (University of Oxford) to KMMW, a McDonnell Scholar Award to JHM, and a

Wellcome Principal Research Fellowship to AJK (WT076508AIA, WT108369/Z/2015/Z), which included an Enhancement Award for JHM.

## Additional information

### Competing interests
Andrew J King: Senior editor, *eLife*. The other authors declare that no competing interests exist.

### Funding

| Funder | Grant reference number | Author |
|---|---|---|
| Wellcome | Principal Research Fellowship WT076508AIA | Andrew J King |
| Wellcome | Enhancement Award | Josh H McDermott Andrew J King |
| James S. McDonnell Foundation | Scholar Award | Josh H McDermott |
| Biotechnology and Biological Sciences Research Council | New Investigator Award (BB/M010929/1) | Kerry MM Walker |
| University Of Oxford | DPAG Early Career Fellowship | Kerry MM Walker |
| Wellcome | Principal Research Fellowship WT108369/Z/2015/Z | Andrew J King |

The funders had no role in study design, data collection and interpretation, or the decision to submit the work for publication.

### Author contributions
Kerry MM Walker, Conceptualization, Software, Formal analysis, Supervision, Funding acquisition, Investigation, Visualization, Methodology, Writing—original draft, Project administration, Writing—review and editing, Acquired data; Ray Gonzalez, Data curation, Software, Investigation, Acquired data; Joe Z Kang, Investigation, Methodology, Acquired data and approved final version for publication; Josh H McDermott, Conceptualization, Resources, Software, Supervision, Funding acquisition, Methodology, Project administration, Writing—review and editing; Andrew J King, Conceptualization, Resources, Supervision, Project administration, Writing—review and editing

### Author ORCIDs
Kerry MM Walker (iD) http://orcid.org/0000-0002-1043-5302
Andrew J King (iD) http://orcid.org/0000-0001-5180-7179

### Ethics
Human subjects: Informed consent was obtained from human participants. Consent to publish was not required, as there is no identifying information present in the manuscript. All experimental procedures on humans were approved by the Committee on the Use of Humans as Experimental Subjects at MIT (Protocol 1208005210).
Animal experimentation: The animal procedures were approved by the University of Oxford Committee on Animal Care and Ethical Review and were carried out under license from the UK Home Office, in accordance with the Animals (Scientific Procedures) Act 1986 and in line with the 3Rs. Project licence PPL 30/3181 and PIL l23DD2122.

### Decision letter and Author response
Decision letter https://doi.org/10.7554/eLife.41626.014
Author response https://doi.org/10.7554/eLife.41626.015

# Additional files

## Supplementary files

• Supplementary file 1. Table 1a-k provide further details of all statistical tests described in the article.
DOI: https://doi.org/10.7554/eLife.41626.009

• Transparent reporting form
DOI: https://doi.org/10.7554/eLife.41626.010

## Data availability

All psychophysical data and stimuli for this study have been uploaded to Dryad (doi:10.5061/dryad.95j80kv).

The following dataset was generated:

| Author(s) | Year | Dataset title | Dataset URL | Database and Identifier |
|---|---|---|---|---|
| Walker KMM, Gonzalez R, Kang JZ, McDermott JH | 2018 | Data from: Pitch perception is adapted to species-specific cochlear filtering | https://doi.org/10.5061/dryad.95j80kv | Dryad Digital Repository, 10.5061/dryad.95j80kv |

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
