## [Decision Letter]

Thank you for submitting your article "Pitch perception is adapted to species-specific cochlear filtering" for consideration by *eLife*. Your article has been reviewed by Eve Marder as the Senior Editor, a Reviewing Editor, and two reviewers. The following individual involved in review of your submission has agreed to reveal his identity: Daniel Bendor (Reviewer #1).

The reviewers have discussed the reviews with one another and the Reviewing Editor has drafted this decision to help you prepare a revised submission.

Summary:

Pitch perception is important, but its neurobiological basis remains unsettled. In this study, the spectral and temporal cues used for pitch discrimination are compared between humans and ferrets. The authors find that unlike humans, ferrets use predominantly temporal cues, relying on higher order harmonics for pitch discrimination. The experiment is well designed, and the data are compelling. With these strengths, there are concerns about novelty, and experimental design.

Essential revisions:

1) The basic conclusions have been drawn in earlier papers. The relationship between broader cochlear tuning and greater reliance on unresolved harmonics has been explicitly stated in Shofner and Chaney, 2013 and are present in other studies of the perception and neural coding of pitch in non-human mammals. We do understand that there are benefits to doing the same experiment in humans and ferrets and suggest increased clarity in covering how the main conclusions have already been drawn by previous studies.

2) The choice of F0s of 500 and 1000 Hz is odd. First, they are exactly an octave apart. Second, do the authors believe that ferrets are sensitive to 1000-Hz temporal modulations? Couldn't this discrimination simply be detecting some modulation at 500 Hz, and no modulation at 1000 Hz, and not have anything to do with pitch? The authors should discuss the caveats of using this stimulus, which still clearly demonstrates that animals are doing something different from the humans. The authors might more explicitly acknowledge this potential experimental weakness.

[Editors' note: further revisions were requested prior to acceptance, as described below.]

Thank you for submitting your article "Pitch perception is adapted to species-specific cochlear filtering" for consideration by *eLife*. Your article has been reviewed by three peer reviewers, one of whom is a member of our Board of Reviewing Editors, and the evaluation has been overseen by Eve Marder as the Senior Editor.

The reviewers have discussed the reviews with one another and the Reviewing Editor has drafted this decision to help you prepare a revised submission.

Essential revisions:

Your reviewers had a split decision about your paper, over both novelty and stimulus design issues. We have come to the consensus that an amended introduction and discussion could potentially resolve our concerns. Please carefully re-evaluate the original critiques, in addition to the points summarized below and the new full reviews.

First, it may be that you were not aware of the other Shofner paper in humans. A more balanced introduction and discussion should include discussion of the pair of previous studies that already compared humans with another similarly-sized mammal, and used similar stimuli in both, and came to the same conclusions.

Second, the stimulus design issues might be resolved for this paper by a more thorough accounting of the available evidence pointing to insensitivity to modulation around 1000 Hz in all species tested so far (with relevant citations added). We suggest you acknowledge the potential confounds remaining in the human version of the experiment, where the spectral edge is not held constant. You might also discuss whether ferrets might be able to perceive higher modulation rates than humans or may only need to perceive the lower pitch to perform the task correctly.

One reviewer points out that it is possible to design an experiment that does not have these confounds. For instance, using shallow bandpass filter slopes so that the spectral edge of the stimulus is not a salient cue, would remove the possibility that the spectral edge is being used. Similarly, using F0s within the range where F0 can actually be detected via envelope cues (at least in humans), would remove concerns about the frequencies you used. After discussion, we do not require these experiments for this paper, but think that discussion of this point would be helpful.

Reviewer #2:

This paper provides an interesting example of how differences in peripheral spectral resolution may influence higher-level percepts, in this case pitch. My primary concerns with the original submission were (1) The findings were not as original as implied by the authors (with previous studies having ascribed differences between human and animal pitch perception and performance to differences in spectral resolution), and (2) It was not clear from the stimuli used whether pitch was in fact the dimension that the ferrets were using to perform the task, particularly as the higher-F0 conditions (between 500 and 1000 Hz) extends beyond the region for which envelope-based pitch had been shown to exist.

Unfortunately, the revision does not provide a satisfactory rebuttal to either concern. The main claim to novelty here is that the same task was used in the two species. Although this is true, the tasks used in previous papers were not all that different. For instance: Shofner and Chaney, (2013) showed in chinchillas that temporal-envelope cues, and not spectral peaks (ie. resolved harmonics) were used for pitch cues, and Shofner and Campbell, (2012 not cited by the current authors) showed in humans, using essentially identical stimuli that the opposite pattern of results was observed. In both papers, the authors explicitly invoke species differences in spectral resolution to explain the difference in results. In my view, this makes the present contribution an incremental improvement – one that deserves publication, but not in a general-audience journal such as *eLife*.

Even if the argument that the same stimuli and task makes the current study worthy of publication, there are also problems on this front: Although the same basic task was used for the two species, some critical differences were present. First, the range of F0s was much lower for the humans than for the ferrets. Second, the F0 difference between the two target tones was much smaller. This was justified because humans have much higher sensitivity to F0 differences. However, this choice results in a potential confound: For the humans, there was a large difference in the lower spectral edge between the two target stimuli in the control (wideband) condition and in the condition with the lower harmonics present: for the lower-F0 tone (180 Hz), the second harmonics was the lowest (360 Hz), whereas for the higher-F0 tone (220 Hz), the first harmonic was lowest present (220 Hz). This provides a large spectral-edge cue that the ferrets did not have access to. Humans had access to this cue in all but the high high-harmonics condition (in which performance was poorer). Although this does negate the results of the study, it is an important confound that undermines the claim to novelty of having used the same stimuli and tasks in the two populations.

The final major point relates to the use of an F0 of 1000 Hz in conditions where only unresolved harmonics should be present. The authors do mention that this is likely above the existence region of pitch in cats but omit to mention that it is also above the existence region of envelope-based periodicity pitch in humans (e.g., Burns and Viemeister, 1976, 1981; Carlyon and Deeks, 2002). Indeed, humans not only hear little or no pitch with 1000 Hz modulation, they are very insensitive to even the presence of modulation at 1000 Hz. As shown by Viemeister, (1979) and by Kohlrausch, Fassel and Dau (2000), sensitivity to amplitude modulation degrades beyond about 150 Hz (around 50 Hz with noise carriers for different reasons), in a way that is independent of CF, and so is unlikely to be due to peripheral filtering. Citing Oxenham et al., (2011) as evidence for F0 perception beyond 1000 Hz misses the point, as that involved resolved harmonics, where envelope cues were unlikely to have been present. If it is not perceived as pitch in either humans or cats, what is the likelihood of it being perceived as pitch in ferrets? Why isn't it possible that ferrets are basing their judgments on changes in perceived modulation strength, rather than pitch itself?

*Reviewer #3:*

This is a revision of a paper about pitch perception. The field is cloudy, and the authors attempted to clarify some controversial issues by comparing spectral and temporal cues for pitch discrimination between humans and ferrets. The authors propose that unlike humans, ferrets use predominantly temporal cues, relying on higher order harmonics for pitch discrimination. i.e. there may be different neurobiological underpinnings for similar pitch discrimination.

The revised version of the paper addressed the two major flaws identified by the reviewers. These were claims of novelty, and weaknesses in experimental design.

In the revised version, the claims of novelty were toned down, and explanations provided for the choice of stimuli, although there remain concerns about whether the stimuli can really be the same for the two species.

---

## [Author Response]

Essential revisions:1) The basic conclusions have been drawn in earlier papers. The relationship between broader cochlear tuning and greater reliance on unresolved harmonics has been explicitly stated in Shofner and Chaney, 2013 and are present in other studies of the perception and neural coding of pitch in non-human mammals. We do understand that there are benefits to doing the same experiment in humans and ferrets and suggest increased clarity in covering how the main conclusions have already been drawn by previous studies. Details are provided in minor comments below.

The reviewers are correct to point out that the idea that cochlear filter widths might relate to the use of resolved harmonics is not a new idea. The unique contribution of our manuscript is, as the Reviewers point out, to demonstrate the species differences predicted by this idea more conclusively than previous papers. We achieved this by testing both humans and a non-human animal on a behavioural task that was (a) specifically designed to probe the use of resolved harmonic and temporal pitch cues, and (b) which could be directly compared across species. Our simulation work builds on previous discussions of cochlear filter widths and their relation to resolved harmonic by providing visualizations of the spectral and temporal representation of harmonic complexes across species that differ in cochlear widths.

We have made edits to our manuscript to clarify our novel contributions and their relation to Shofner and Chaney, (2013).

2) The choice of F0s of 500 and 1000 Hz is odd. First, they are exactly an octave apart. Second, do the authors believe that ferrets are sensitive to 1000-Hz temporal modulations? Couldn't this discrimination simply be detecting some modulation at 500 Hz, and no modulation at 1000 Hz, and not have anything to do with pitch? The authors should discuss the caveats of using this stimulus, which still clearly demonstrates that animals are doing something different from the humans. The authors might more explicitly acknowledge this potential experimental weakness.

Our initial F0s of 500 and 1000Hz were intentionally chosen to be an octave apart. This allowed us to exactly match the frequency bandwidths of stimuli with different F0s (by putting the lower and upper cutoff frequencies at a common harmonic). Otherwise, ferrets could have used the spectral edges of stimuli, rather than F0, to discriminate between them. For the same reason, our second set of testing stimuli (150 and 450Hz), in which we replicated all our behavioural findings (Figure 5A), were separated by a factor of three. The reviewer’s concern with the octave spacing may be that the task would be slightly more difficult with octave spacings due to similarities in pitch chroma (Yin, Fritz and Shamma, 2010). However, our ferrets learned to discriminate both sets of F0s well, and in fact even performed better on the one-octave spacing (500, 1000 Hz) than factor-of-three spacing (150, 450 Hz).

To clarify the motivations for our choice of target stimuli, we have added the following sentence to subsection “Testing stages and stimuli”:

“The pairs of target stimuli were chosen to be either 1 octave (a factor of two; 500 and 1000Hz) or a factor of three (150 and 450Hz) apart so that their ranges of harmonics could be matched exactly in spectral range.”

And to subsection “Behavioural measures of pitch cue use in ferrets”:

“In both cases, the integer ratios between the F0s to be discriminated allowed us to match their spectral bandwidths exactly, so that ferrets could not solve the task based on the frequency range of the sound (Figure 3; left column)”

In relation to the reviewer’s second point, we fully expect that ferrets should be sensitive to a 1000Hz modulation. This is within the limits of strong phase locking reported in recordings of the auditory nerve in ferrets (Sumner and Palmer, 2012) and cats (Johnson, 1980), and well within the upper pitch limit of ~3500Hz for human listeners (Ritsma, 1962; Oxenham et al., 2011). It is also plausible that ferrets would be more sensitive to fast modulations than human listeners given their apparent importance in ferret pitch perception. The Reviewers’ concern may be based on the study of Chung and Colavita, (1976), who demonstrated periodicity pitch perception in cats on a transfer of learning task up to F0s of 800Hz. However, the upper “pitch” limit in that study could actually reflect the cats’ limits in generalizing across very different spectral ranges, which were higher for higher F0s in that study.

We also note that our pattern of results was replicated when we trained ferrets to discriminate between 150 and 450 Hz F0s, which are within a range where temporal modulation rate discrimination should be good even in humans. The consistency of results across these two very different F0 ranges suggests that the animals are using the cues in the same way in both cases.

To address the reviewer’s concern about the 1000Hz F0, we have added the following new paragraph to subsection “Implications for neurophysiological work”:

“It is interesting that ferrets appear to favour temporal cues over resolved harmonic cues even for F0s as high as 1000Hz. This periodicity is well below the upper limits of human pitch perception (Oxenham et al., 2011) and the phase locking limit of the ferret auditory nerve (Sumner and Palmer, 2012), though it falls above a behavioural measurement of periodicity pitch limits in cats (Chung and Colavita, 1976). While we cannot state conclusively that ferrets perceived the 1000Hz modulation, the similar pattern of performance for the lower pair of reference F0s (150 and 450 Hz) suggests that the ferrets were using the same strategies for all F0s tested.”

[Editors' note: further revisions were requested prior to acceptance, as described below.]

Essential revisions:Your reviewers had a split decision about your paper, over both novelty and stimulus design issues. We have come to the consensus that an amended introduction and discussion could potentially resolve our concerns. Please carefully re-evaluate the original critiques, in addition to the points summarized below and the new full reviews.

We have revised the Introduction and Discussion section as requested. These revisions are summarized in the responses below.

In light of the reviewer’s feedback and our changes to the manuscript during the review process, we have changed the Title of the manuscript from the original:

“Pitch perception is adapted to species-specific cochlear filtering”

to:

“Across-species differences in pitch perception are consistent with differences in cochlear filtering”

1) First, it may be that you were not aware of the other Shofner paper in humans. A more balanced introduction and discussion should include discussion of the pair of previous studies that already compared humans with another similarly-sized mammal, and used similar stimuli in both, and came to the same conclusions.

We were indeed unaware of the study by Shofner and Campbell, (2012), and we thank the reviewer for suggesting it to us. We now cite this work in our discussion of the human literature.

(Introduction) “Although psychophysical experiments have demonstrated that humans can extract F0 using either resolved harmonics or unresolved harmonics alone (Bernstein and Oxenham, 2003; Houtsma and Smurzynski, 1990; Shackleton and Carlyon, 1994), human pitch perception is generally dominated by resolved harmonics (Ritsma, 1967; Shackleton and Carlyon, 1994; Shofner and Campbell, 2012).”

We have also pointed out in the revised Introduction that Shofner and colleagues explicitly compared the patterns of performance between humans and chinchillas, though we note that the tasks used were quite different between the two species. Chinchillas were required to report when the standard sound (1-channel harmonic tone complex; wHTC) changed to a test sound (a tonal HTC or a HCT with varied numbers of vocoded channels), in a classic go/no-go paradigm. Animals could equally respond on this task to a change in pitch or in some other timbre dimension, with this design determining the threshold for detecting a salient pitch change. Humans, by contrast, were required to judge the pitch salience of a test sound (the HCT with varied numbers of vocoded channels) on a percentage scale, where 0% matched a standard wHTC presented at the start of the trial and 100% matched a clean HTC presented at the end of the trial. Given these considerable task differences, we would argue that our current direct comparison of human and animal performance on the same pitch discrimination task remains a unique and valuable contribution to the field. This is now included in our Introduction:

“Finally, most animal studies have not directly compared performance across human and non-human species (Bendor et al., 2012; Osmanski et al., 2013; Song et al., 2016), or have compared them across considerably different behavioural tasks (e.g. Shofner and Campbell, 2012 versus Shofner and Chaney, 2013), so differences in task demands might account for any apparent species differences.”

We also emphasize in the Introduction that the relative widths of cochlear filters has been offered as a explanation for the greater reliance of animals on temporal pitch cues in discussions of these studies by Shofner and colleagues, as is also the case in the work of Wang and colleagues. Our manuscript builds on this suggestion by demonstrating how these filter widths would manifest as activation profiles in the modelled human and ferret auditory nerve. This previously existing contribution is directly acknowledged in the Introduction:

(Introduction) “It has been suggested that these apparent species differences in perception could relate to the pitch cues that are available following cochlear filtering (Cedolin and Delgutte, 2010; Shofner and Chaney, 2013).”

Similarly, we have stated that previous authors have described narrower filter widths in ferrets:

(Introduction) “In particular, the growing evidence that cochlear bandwidths are broader in many other species (Joris et al., 2011; Shera et al., 2002), including ferrets (Alves-Pinto et al., 2016; Sumner et al., 2018), supports the possibility that they might process pitch cues in different ways from humans, as has been noted (Shofner and Campbell, 2012; Shofner and Chaney, 2013).”

We reiterate this point elsewhere in our manuscript:

(Subsection “Findings in other species”) “Our cochlear simulations suggest that harmonic resolvability is worse for ferrets than human listeners, as a consequence of their previously demonstrated differences in cochlear filter widths (Sumner et al., 2018).”

The revised Discussion section also points out that Shofner previously proposed that species differences in pitch perception could be explained by differences in peripheral tuning:

(Subsection “Findings in other species”) “Chinchillas were similarly shown to detect the onset of a periodic sound following a non-periodic sound using temporal, rather than resolved harmonic, cues (Shofner, 2002). While these studies did not explicitly compare the use of resolved and unresolved pitch cues, their findings are consistent with ours regarding the importance of temporal cues in non-human species, and the authors similarly proposed that species differences could be explained by differences in peripheral tuning.”

2) Second, the stimulus design issues might be resolved for this paper by a more thorough accounting of the available evidence pointing to insensitivity to modulation around 1000 Hz in all species tested so far (with relevant citations added). … You might also discuss whether ferrets might be able to perceive higher modulation rates than humans or may only need to perceive the lower pitch to perform the task correctly.

We agree that a discussion of the upper limit of periodicity pitch in human listeners would be a useful addition and have added it to the revised discussion. This limit was estimated by Carlyon and Deeks, (2002) and later more thoroughly by Macherey and Carlyon, (2014) to be potentially as high as 700-800Hz. It is unknown if this same upper limit applies to other species and given the greater reliance on temporal pitch cues that has now been reported for all non-human animals studied (ferrets, marmosets, gerbils and chinchillas), a higher temporal pitch limit in these species remains entirely possible. We now include a citation of this work and clarify that the upper limit of temporal pitch in humans is likely to be below 1000Hz.

We agree that it is possible that ferrets are basing their judgments in the 707Hz version of our task by detecting the presence of modulation in the 500Hz target and absence in 1000Hz target. We have added to our text to explicitly acknowledge this possibility.

The new text incorporating the above suggestions now reads:

(Subsection “Implications for neurophysiological work”) “It is interesting that ferrets appear to favour temporal cues over resolved harmonic cues even for F0s as high as 1000Hz. While this periodicity is well below the phase locking limit of the ferret auditory nerve (Sumner and Palmer, 2012), it falls above a reported temporal pitch limit in cats (Chung and Colavita, 1976) and humans (Carlyon and Deeks, 2002; Macherey and Carlyon, 2014). We thus cannot rule out the possibility that ferrets were actually performing the 707Hz reference task by detecting the absence of temporal modulation at 1000Hz. However, the similar pattern of performance for the lower pair of reference F0s (150 and 450 Hz) is less plausibly explained in this way, and supports the idea that the ferrets were basing their judgments on temporal pitch cues. It is also possible that the temporal modulation limit of pitch is higher in an animal that relies more on this cue.”

3) We suggest you acknowledge the potential confounds remaining in the human version of the experiment, where the spectral edge is not held constant. One reviewer points out that it is possible to design an experiment that does not have these confounds. For instance, using shallow bandpass filter slopes so that the spectral edge of the stimulus is not a salient cue, would remove the possibility that the spectral edge is being used. … After discussion, we do not require these experiments for this paper, but think that discussion of this point would be helpful.

It is correct that the lower spectral edge of the stimuli in the “low harmonics” condition differed for the low and high F0 targets for human subjects, and that this difference is unavoidable due to the small F0 difference required to test human pitch performance. However, if subjects did choose to make pitch judgments based on the position of the lower spectral edge of probe stimuli, the presence of this cue would likely impair their performance rather than improve it, as the higher F0 (220 Hz) had a lower spectral edge (220 Hz) and the lower F0 (180 Hz) a higher one (360 Hz). Listeners were given no feedback on probe trials, so it is unlikely that they would have learned the counterintuitive strategy of reporting the stimulus with the lower spectral edge as having the higher F0. Thus, if anything, this issue would have most likely impaired human performance on the low harmonics condition relative to ferrets, when in fact this condition is the one where humans are better than ferrets. This confound is thus conservative with respect to the results and conclusions.

The existence of these spectral edge cues is evident from our stimulus descriptions (e.g. Figure 4B). We have modified our text to emphasize the existence of these spectral edge cues, and to explain why they are unlikely to aid performance on our task.

In the Results section:

“The constraints of using a smaller F0 difference necessitated a different lowest frequency component between the two F0s for the “All Harmonic” and “Low Harmonic” conditions, unlike the ferret stimuli. However, the spectral edge was higher for the lower F0 (360 Hz vs. 220 Hz), and thus is unlikely to have provided a cue that could be used to correctly perform the task (particularly given that we did not provide feedback). Because the F0 difference differed between species, the between-species comparison of interest here is not the difference in absolute scores on the task, but the pattern of performance across probe conditions.”

In the Materials and methods section:

“Human subjects were tested on a pitch classification task that was designed to be as similar as possible to Stage 3 of ferrets’ task (see above). Target F0s of 180 and 220 Hz were tested on 16 subjects. Due to the smaller F0 difference required to make the task difficult enough to challenge human listeners (Walker et al., 2009), it was not possible to match the lower spectral edge of the “Low Harmonic” and “All Harmonic” stimuli as we did for ferrets. However, the stimuli were set such that the higher F0 target had the lower spectral edge. As a result, these edge cues were incongruent with the F0. Because feedback was not provided, it is unlikely that subjects could have learned to associate a lower spectral edge with the higher F0 and vice versa. This stimulus confound thus if anything is likely to have made the task more difficult in the “Low Harmonic” and “All Harmonic” conditions. Because our main finding is that the relative performance of humans was better than that of ferrets in these conditions, it seems unlikely to have influenced the key results.”

4) Using F0s within the range where F0 can actually be detected via envelope cues (at least in humans), would remove concerns about the frequencies you used. After discussion, we do not require these experiments for this paper, but think that discussion of this point would be helpful.

Our experiments in fact already include F0s within this range: the 150Hz vs 450Hz discrimination task for ferrets and the 180Hz vs 220Hz task for humans. All of these are within the reported range of periodicity pitch for humans (Macherey and Carlyon, 2014). Ferrets perform very poorly in discriminating lower F0 pitches (Walker et al., 2009), so testing even lower F0 ranges is not feasible. Keep in mind that the lower limit of the ferret hearing range is higher than that of human listeners, so very low pitches may be less relevant for them than they are for us.

To make it more obvious that ferrets were presented with stimuli from this range as well as from the higher range, we now depict both in Figure 3.

We also now draw the reader’s attention to the fact that we have included a low F0 range for ferrets and find exactly the same pattern of results as we did for the high F0 range in our manuscript:

(subsection “Implications for neurophysiological work”) “We thus cannot rule out the possibility that ferrets were actually performing the 707Hz reference task by detecting the absence of temporal modulation at 1000Hz. However, the similar pattern of performance for the lower pair of reference F0s (150 and 450 Hz) is less plausibly explained in this way and supports the idea that the ferrets were basing their judgments on temporal pitch cues.”

Reviewer #2:This paper provides an interesting example of how differences in peripheral spectral resolution may influence higher-level percepts, in this case pitch. My primary concerns with the original submission were (1) The findings were not as original as implied by the authors (with previous studies having ascribed differences between human and animal pitch perception and performance to differences in spectral resolution), and (2) It was not clear from the stimuli used whether pitch was in fact the dimension that the ferrets were using to perform the task, particularly as the higher-F0 conditions (between 500 and 1000 Hz) extends beyond the region for which envelope-based pitch had been shown to exist.1) The main claim to novelty here is that the same task was used in the two species. Although this is true, the tasks used in previous papers were not all that different. For instance: Shofner and Chaney, (2013) showed in chinchillas that temporal-envelope cues, and not spectral peaks (ie resolved harmonics) were used for pitch cues, and Shofner and Campbell, (2012 not cited by the current authors) showed in humans, using essentially identical stimuli that the opposite pattern of results was observed. In both papers, the authors explicitly invoke species differences in spectral resolution to explain the difference in results. In my view, this makes the present contribution an incremental improvement – one that deserves publication, but not in a general-audience journal such as eLife.

Please see our reply this concern in our response to point 1, above.

2) First, the range of F0s was much lower for the humans than for the ferrets.

Humans were tested on target F0s of 180 and 220Hz, which is in fact within the 150 and 450 Hz target range tested in ferrets. However, ferrets were additionally tested on a higher F0 range (500 and 1000Hz). We included this additional higher F0 range for ferrets because their pitch discrimination thresholds are poorer for low F0s (Walker et al., 2009). Therefore, the low F0 range was matched in humans and ferrets in an acoustical sense, while the higher F0 range provided a better match in terms of F0 discrimination preference. The same pattern of results was observed for both the low and high ranges tested in ferrets, demonstrating that they are independent of the absolute F0 used.

We suspect the reviewer may not have noticed the lower F0s used for the ferret. To make the two sets of stimuli that were used more obvious, we now depict both of them in Figure 3.

We also added this justification for the two F0 ranges to the methods:

(Subsection “Testing stages and stimuli”) “The 150 and 450 Hz targets were chosen to overlap with the F0 range that we tested in human listeners (below). The 500 and 1000 Hz condition was included as ferrets often perform better on pitch discrimination tasks in this range than on sounds with lower F0s (Walker et al., 2009).”

3) Second, the F0 difference between the two target tones was much smaller.

We were unable to match the exact F0’s presented to humans and ferrets in our pitch discrimination task. Using the same F0’s would result in the task being too difficult for ferrets to perform with even the standard stimulus, or too easy for humans for even the most degraded pitch cues (see Walker et al., 2009). We argue that it is more important to match the difficulty of our task than the absolute F0s presented. We have clarified this choice of stimuli in our Results, in the following text:

(Subsection “Comparison of human and ferret pitch classification performance”) “We tested human listeners using the same types of standard and probe stimuli as in the final stage of ferret testing described above. As the pitch discrimination thresholds of human listeners are known to be superior to those of ferrets (Walker et al., 2009), we adapted the target F0s (180 and 220 Hz) and harmonic cut-offs for human hearing (Figure 4). The constraints of using a smaller F0 difference necessitated a different lowest frequency component between the two F0s for the “All Harmonic” and “Low Harmonic” conditions, unlike the ferret stimuli. However, the spectral edge was higher for the lower F0 (360 Hz vs. 220 Hz), and thus is unlikely to have provided a cue that could be used to correctly perform the task (particularly given that we did not provide feedback). Because the F0 difference differed between species, the between-species comparison of interest here is not the difference in absolute scores on the task, but the pattern of performance across probe conditions.”

4) For the humans, there was a large difference in the lower spectral edge between the two target stimuli in the control (wideband) condition and in the condition with the lower harmonics present: for the lower-F0 tone (180 Hz), the second harmonics was the lowest (360 Hz), whereas for the higher-F0 tone (220 Hz), the first harmonic was lowest present (220 Hz). This provides a large spectral-edge cue that the ferrets did not have access to. Humans had access to this cue in all but the high high-harmonics condition (in which performance was poorer). Although this does negate the results of the study, it is an important confound that undermines the claim to novelty of having used the same stimuli and tasks in the two populations.

Please see our reply to this concern in our response to point 3, above.

5) The final major point relates to the use of an F0 of 1000 Hz in conditions where only unresolved harmonics should be present. The authors do mention that this is likely above the existence region of pitch in cats but omit to mention that it is also above the existence region of envelope-based periodicity pitch in humans (e.g., Burns and Viemeister, 1976, 1981; Carlyon and Deeks, 2002). Indeed, humans not only hear little or no pitch with 1000 Hz modulation, they are very insensitive to even the presence of modulation at 1000 Hz. As shown by Viemeister, (1979) and by Kohlrausch, Fassel and Dau (2000), sensitivity to amplitude modulation degrades beyond about 150 Hz (around 50 Hz with noise carriers for different reasons), in a way that is independent of CF, and so is unlikely to be due to peripheral filtering. Citing Oxenham et al., (2011) as evidence for F0 perception beyond 1000 Hz misses the point, as that involved resolved harmonics, where envelope cues were unlikely to have been present. If it is not perceived as pitch in either humans or cats, what is the likelihood of it being perceived as pitch in ferrets? Why isn't it possible that ferrets are basing their judgments on changes in perceived modulation strength, rather than pitch itself?

We take the point and have revised the manuscript to acknowledge this issue. Our modifications to the manuscript that address this issue are described above, under points 2 and 4.

In addition to the concerns summarized by the editor, the reviewer points out here that our manuscript cited Oxenham, 2011 as evidence for the upper pitch limit, not the upper limit of temporal pitch. As the reviewer has pointed out that this could be confused with an upper limit of temporal pitch, we have removed the citation and accompanying text from our manuscript and replaced it with a citation of Carlyon’s work:

(Subsection “Implications for neurophysiological work”) “While this periodicity is well below the phase locking limit of the ferret auditory nerve (Sumner and Palmer, 2012), it falls above a reported temporal pitch limit in cats (Chung and Colavita, 1976) and humans (Carlyon and Deeks, 2002; Macherey and Carlyon, 2014).”

Reviewer #3:This is a revision of a paper about pitch perception. The field is cloudy, and the authors attempted to clarify some controversial issues by comparing spectral and temporal cues for pitch discrimination between humans and ferrets. The authors propose that unlike humans, ferrets use predominantly temporal cues, relying on higher order harmonics for pitch discrimination. i.e. there may be different neurobiological underpinnings for similar pitch discrimination.The revised version of the paper addressed the two major flaws identified by the reviewers. These were claims of novelty, and weaknesses in experimental design.In the revised version, the claims of novelty were toned down, and explanations provided for the choice of stimuli, although there remain concerns about whether the stimuli can really be the same for the two species.

We thank the reviewer for recognizing the importance of our manuscript. We have addressed the concerns about matching stimuli and tasks across species above, in our responses to 3-4, and reviewer 2’s points 2 and 3.